# A Non-Geodesic Trajectory Design Method and Its Post-Processing for Robotic Filament Winding of Composite Tee Pipes

**DOI:** 10.3390/ma14040847

**Published:** 2021-02-10

**Authors:** Cheng Chang, Zhenyu Han, Xinyu Li, Shouzheng Sun, Jihao Qin, Hongya Fu

**Affiliations:** School of Mechatronics Engineering, Harbin Institute of Technology, No. 92, Xidazhi Street, Harbin 150001, China; 19S008117@stu.hit.edu.cn (C.C.); hanzy@hit.edu.cn (Z.H.); 20S108276@stu.hit.edu.cn (X.L.); 17S008134@stu.hit.edu.cn (J.Q.); hongyafu@hit.edu.cn (H.F.)

**Keywords:** polymer-matrix composite, tee pipe, trajectory design, post-processing, robotic filament winding

## Abstract

With the advantages of high specific strength and well corrosion resistance, polymer-matrix composite tee pipes are widely used in aerospace and civilian fields. The robotic filament winding technology is suitable for forming complex shape parts. This paper aims to provide a novel non-geodesic trajectory design method to get a continuous trajectory for tee pipe winding. Furthermore, post-processing methods are proposed for realizing the full coverage of tee pipes by robotic filament winding. The CAD/CAM software is then designed to simulate the winding process and realize the cover of the whole tee pipe. Finally, experiments of winding a tee pipe with a desktop winding machine and a six-axis winding robot are carried out. The results show that the tee pipe is fully covered, verifying the accuracy of the design method and post-processing methods.

## 1. Introduction

Fiber-reinforced resin composites have the characteristics of high specific strength, large specific modulus, fatigue resistance, corrosion resistance, heat resistance, and low thermal expansion coefficient so that they have always drawn wide attention in many industrial areas [1,2,3,4,5]. Tee pipes are commonly used pipe connectors and support connection parts, made from polyethylene, polypropylene, alloy steel, glass fiber reinforced plastic, etc. Compared with pipes made from other materials, fiber-reinforced resin composite tee pipes have the advantages of both high specific stiffness and good corrosion resistance, meanwhile, their thermal insulation performance is good, and the surface roughness is low [6,7,8,9].

There are various molding methods for resin-based fiber-reinforced composites, including hand lay-up, fiber placement, RTM (resin transfer molding), fiber winding, etc. The characteristics and applications of these processes are different. Among those methods, RTM and fiber winding are suitable for tee pipes manufacturing. In the RTM process, tee pipes are formed in a closed mold, with good resin wettability, smooth double surfaces, and high forming efficiency [10,11]. R. Fangueiro et al. successfully used the RTM process to prepare three-way pipes [12], but the RTM equipment cost is too high. Compared with RTM, fiber winding with advantages of low cost and high efficiency is more suitable for producing tee pipes with good mechanical properties [13,14,15]. Moreover, as the industrial robot develops, robotic filament winding technology with high degrees of freedom is applied to manufacture complex shape parts [16].

However, currently, the molding method of composite tee pipes mainly relies on manual winding and manual cladding. The reason is that the shape of tee pipes is complicated, of which the straight pipe is perpendicular to the main pipe, and the intersection area is a concave curved surface. Therefore, to accomplish the automation of winding, it is studied that the special method of trajectory design for tee pipes. Filament winding trajectory design can be divided into two aspects: local trajectory design and full coverage analysis.

The local trajectory design is responsible for the stability and non-overhead of the winding trajectory. In terms of the trajectory design of tee pipes, Sanjeev Seereeram et al. set up hanging nails at the opening of the tee pipe and designed a winding trajectory based on the geodesic for the tee pipe [17]. Friedrich. Ralph S et al. designed 11 winding trajectories for the tee pipe, but did not mention the design method of each trajectory, and only gave a sketch of the trajectory shape [18]. Zhenyu Han et al. discussed the conditions satisfied by the winding angle at the intersection of the tee pipe [19]. Yuefeng Wei designed a variety of winding trajectories for a smooth transition and direct intersecting three-way pipes and realized the function of generating a trajectory between two designated points by adjusting the winding angle multiple times so that the winding trajectory has a degree of designability [20]. For the non-geodesic filament winding process, Wang R et al. studied the slippage coefficient measurement for different mandrel shapes [21]. Marius Dackweiler et al. came up with an analytical model of the geometry of a T-joint as well as a differential-geometric approach combined with an algorithm to calculate the winding paths for joining different profiles. But the method focused on the T-joint for connection with profiles and the winding paths cannot be applied for winding a whole tee pipe. Because the paths haven’t taken the trajectory on the straight pipes and the movement of the winding machine which should avoid the interferences in the winding process in consideration [22].

The global coverage analysis ensures that the fibers can be evenly distributed on the mold, which is of great engineering significance. At present, there are few studies on global coverage at home and abroad, especially on special-shaped parts. Xianfeng Wang divided the intersecting part of the main pipe and the branch pipe into three zones and analyzed the direction of the winding trajectory and the number of fiber bundles for each zone, through which the transition part is covered [23]. In the straight pipe part, it is covered with a wraparound method. However, it did not discuss the trajectory connection among the various parts of the core mold, and the crossover between the lines is too small, and the mechanical performance is affected. The trajectory designed by Yuefeng Wei mostly covered the tee pipe, but because there is no bandwidth display, whether there is an area that is not yet covered is unknown [20].

It is still an unsolved problem in the existing fiber winding theory to find the continuous fiber filling method of tee pipes mathematically. The winding of tee pipes has not yet been completely solved. Most of the existing work stays in the application of the general winding theory of the tee pipe and does not specifically analyze the winding characteristics of the tee pipe. Additionally, in the winding process of tee pipes, there are many interference situations. Existing work does not thoroughly discuss the causes and solutions of interferences.

As for winding CAD/CAM technology which is closely connected with winding theory, there have been many mature applications of commercial CAD/CAM software, such as Cadwind, Cadfil, Composicad and Winding Expert, which already have rich functions after so many years’ development. For example, Cadwind can generate winding trajectories for straight pipes, elbows, tee pipes and rotary components. In terms of post-processing, Cadwind can output motion control programs for different winding machines, including 2 to 6-axis winding machines, three-way pipe winding machines and robots. It also supports modeling of the friction between the fiber and the mold, and non-geometric winding. Cadwind can display solid modeling of the wound fiber layer and export relevant data for finite element analysis [24].

The FW Simulation developed by FU and others at Pusan University has done detailed work on the visualization of fiber width, and proposed a calculation method for the boundary line of the fiber, which solved the problem of mutual covering between the fiber and the mold’s surface in the OpenGL development environment [25]. In terms of bending pipe winding, Haisheng Li of Zhejiang University developed the ElbowCAD system. It is developed by VC++ and graphics function library OpenGL, including four modules of fiber path planning, machine path generation, motion simulation and stress analysis [26]. In terms of robot winding, Dong Sun of Harbin University of Science and Technology developed the CAD/CAM system integrating the design and manufacture of complex special-shaped parts winding using robot winding [27].

Except the commercial CAD/CAM software, the other software has problems that the software can only show a basic line type on the model instead of the whole winding process, which is not suitable for winding complex-shaped parts like tee pipes and the boundary between fibers is not clear. Compared with the developed commercial CAD/CAM software, the software FiberStudio in this paper emphasizes more on the display of the winding process and interaction between designers and the software to finish the task of fully covering tee pipes.

In response to the above problems of winding tee pipes, this paper proposes a special non-geodesic trajectory design method for tee pipe winding to get a continuous trajectory to fully cover tee pipes. In the post-processing, the yarn exit position coordinates are calculated based on the envelop surface while the interference problems in different situations in the winding process are analyzed and solved. Based on the above, a CAD/CAM software is developed for designing the winding trajectory of tee pipes to fully cover tee pipes, which is achieved by picking up design points on the uncovered areas. Additionally, through the three-dimensional display of the model of tee pipes, fibers, the yarn exit roller and winding process simulation, the interference avoidance is checked. Experiments of winding a tee pipe by a desktop winding machine and a six-axis robot with a winding head are carried out to prove the feasibility and correctness of the trajectory design method and post-processing method.

## 2. Winding Trajectory Design Method of Tee Pipes

This paper adopts the design method of “scheduling parts first and finally connecting all”, that is, designing multiple continuous tracks first, and then connecting them into a complete continuous track. The advantage of it is there is no dependency among tracks before the connection, and modifying one track does not influence the rest. The purpose of this section is to study the specific implementation method of that, including the track design method and trajectory connection algorithm on single and multiple surfaces. Finally, through analyzing the designed trajectories, the full cover of tee pipes is ensured.

### 2.1. Trajectory Design on Different Parts of Tee Pipes

#### 2.1.1. Concept of Design Patches

In order to simplify the design process of winding tee pipes, this paper introduces the concept of design patches. Design patches are the minimum units in the winding trajectory design process, which consist of surfaces, boundaries, and point sets, as shown in Figure 1c. This paper divides tee pipes into two kinds of design patches, straight pipe patches, and T junction patches. Straight pipe patches are cylinders and T junction patches are made up of two tori, two planes, and a semi-cylinder, as shown in Figure 1b. Through that, the trajectory design of tee pipes is converted to track design on a few basic design patches.

T junction patches are symmetric along XY and YZ planes, so every designed track corresponds with three other tracks, as shown in Figure 1d. As the T-junction is symmetric the set of design points on the boundary of the cylinder segment is also symmetric along the corresponding plane. On the straight pipes, the track has two intersection points with boundaries while it must have a return point, of which the tangential direction is perpendicular to the axis of the straight pipe.

There are two common methods of surface trajectory generation: the parametric method and the mesh method. The design patches of tee pipes include torus, planes, and cylinders, which are simple patches, so this paper adopts the parametric method to generate winding trajectories. In order to improve the degree of freedom, non-geodesic is chosen to design winding trajectories. The stability can be ensured as long as the slip force does not exceed the friction. The lateral slip force and positive pressure can be represented by geodesic curvature *k_g_* and normal curvature *k_n_* respectively. In this paper, the slip line coefficient *λ* is used to represent the stability of trajectories, as shown in Equation (1). As long as the slip coefficient of trajectories *λ* is less than the friction coefficient *μ*, fibers remain stable on the designed trajectories.
(1)$$\lambda =\frac{k_{g}}{k_{n}}$$

The differential equations of the non-geodesic trajectory on surfaces are as shown in Equation (2)
(2)$$\left\{\begin{array}{l} \frac{d\alpha }{ds}=\lambda \frac{Ld^{2}u+2Mdudv+Nd^{2}v}{Ed^{2}u+2Fdudv+Gd^{2}v}+\frac{\cos\alpha }{2\sqrt{G}}\frac{\partial \ln E}{\partial v}+\frac{\sin\alpha }{2\sqrt{E}}\frac{\partial \ln G}{\partial u}\\ \frac{du}{ds}=\frac{\cos\alpha }{\sqrt{E}}\\ \frac{dv}{ds}=\frac{\sin\alpha }{\sqrt{G}} \end{array}\right.$$
where *α* is the intersection angle between the curve and the *u* parameter curve of the surface and  *λ* is the slip line coefficient.

#### 2.1.2. Trajectory Generation Algorithm on Patches

This section is about the trajectory generation algorithm on the three different patches of tee pipes. The coordinate system of torus patches is established, as shown in Figure 2a. *R* and *r* define the dimensions of the torus. P is a point on the torus, and its parameter coordinates are *u* and *v*. The non-geodesic differential equations are as shown in Equation (3).
(3)$$\left\{\begin{array}{l} \frac{d\alpha }{ds}=\lambda \left(-\frac{\cos v\;\cos^{2}\alpha }{R+r\cos v}-\frac{\sin^{2}\alpha }{r}\right)-\frac{\sin v\cos\alpha }{R+r\cos v}\\ \frac{du}{ds}=\frac{\cos\alpha }{R+r\cos v}\\ \frac{dv}{ds}=\frac{\sin\alpha }{r} \end{array}\right.$$

In most of the references, the non-geodesic differential equations are all dependent on the surface parameter *u*, which can be obtained by eliminating *s* in Equation (3), as shown in Equation (4).
(4)$$\left\{\begin{array}{l} \frac{d\alpha }{du}=\lambda \left(-\cos v\cos\alpha -\frac{\sin^{2}\alpha \left(R+r\cos v\right)}{r\cos\alpha }\right)-\sin v\\ \frac{dv}{du}=\frac{R+r\cos v}{r}\tan\alpha \end{array}\right.$$

Next, these two equation sets are analyzed through one calculation test to conclude their advantages and disadvantages. In the test, the parameters of the torus are *R* = 100, *r* = 20 while the parameter coordinates of the non-geodesic starting point are u = π/4, V = π, and the sliding line coefficient *λ* = 0.1. Through the four-stage Runge-Kutta iterative method, the parametric coordinates are calculated. Then the winding trajectory is drawn, as shown in Figure 2c,d.

As can be seen in Figure 2c, when the trajectory is about to retrace, it oscillates. This is because the 1/(cosα) and tanα terms in Equation (4) become infinite as α approaches *π/2*, and the numerical calculations no longer converge. In Figure 2d, the numerical value of the differential equations with arc length parameter *s* as the independent variable does not oscillate and backtracks normally. Therefore, the differential equations with arc length parameter *s* as the independent variable are used in the solution of all non-geodesics on surfaces in this paper.

The coordinate system of cylinder patches is shown in Figure 2b. The differential equations of arc length parameter *s* are as shown in Equation (5). The non-geodesic winding trajectory on the cylinder can be obtained by using the fourth-stage Runge-Kutta method.
(5)$$\left\{\begin{array}{l} \frac{d\alpha }{ds}=-\frac{\lambda \;\cos^{2}\alpha }{R}\\ \frac{du}{ds}=\frac{\cos\alpha }{R}\\ \frac{dv}{ds}=\sin\alpha \end{array}\right.$$

The normal curvature of all plane curves is zero. If the fiber is put along the track with the normal curvature of zero, there is no positive pressure between the fiber and the mold. So that there is no friction which can offset the non-geodesic sideslip force. Therefore, non-geodesics in a plane are unstable and not suitable for winding trajectories. In conclusion, the winding trajectory on the plane is a straight line determined by the starting point and the starting direction.

#### 2.1.3. Trajectory Generation Algorithm on T-Junction and Straight Pipes

The T-junction is formed by three design patches, in which all ends of the winding trajectory are on the three boundaries as shown in Figure 3. The trajectories span multiple patches, so it is necessary to synthesize the algorithm on different patches to generate the complete trajectories on the T-junction. The trajectory algorithm spanning different patches is based on the topological relationship among patches to decide the order of using algorithms.

The trajectory generation algorithm of T-junction needs to call the trajectory generation algorithm of the related patches in a certain order according to the topological relationship among patches. The input of the algorithm is the 3D point and the winding direction. First, the trajectory algorithm of the patch where the 3D point is located is called to generate the trajectory on the current patch. When the trajectory reaches the boundary of the patch, the boundary is used as an index to query the next boundary. If the value of the next boundary is not NULL, the track can get the next patch base on the name of the next boundary. Then, the endpoint and the tangential direction at the endpoint are used as the new design point and winding direction, and the winding track is generated on the next patch, and this process is repeated until the boundary of the T-junction is reached. At this time, only the unidirectional trajectory generation of the initial design point is completed. Then reverse the initial winding direction and repeat the previous calculation process to get the trajectory in another direction. The linear trajectories can be combined to obtain a complete winding trajectory on the T-junction.

The winding trajectories generated by laying design points on the T-junction form new design points on the boundary of the T-junction. The winding trajectories on straight pipes are generated according to these design points.

### 2.2. Conjunction of Trajectories on Different Patches

After getting the winding trajectories on different patches, they need to be connected to form a complete continuous track. Both end points of a complete trajectory on the branch pipe should be on the boundary. The algorithm in Section 2.1 can only generate the algorithm between the starting point and the reentry point, and the reentry points need to be connected to obtain the complete trajectory.

The number of trace connection modes is analyzed below. Not any two trajectories can be connected to each other. Only trajectories in opposite directions can be connected to form a complete trajectory. The tee pipe has two symmetric surfaces, and the winding order is expected to be symmetrical so that a neat grid pattern can be formed on the same layer. Therefore, in order to make trajectories symmetrical and neat, trajectories should be connected as early as possible in the design process.

The basic requirement for winding trajectories is that they are continuous and noncircular. The connection algorithm is aimed to connect discrete trajectories into a continuous trajectory, but loops cannot be introduced into the connection process.

Figure 4 shows the trajectories obtained by connecting the trajectories. In the connection algorithm, the point closest to the given trajectory is chosen as the connection point in order to reduce the unevenness of the winding thickness. In this paper, the plane farthest from the boundary of straight pipes is taken as the plane of the turning point. Turning points all retrace on this plane, and their *α* is 90°. The fiber accumulation is serious at turning points. Therefore, in order to reduce the unevenness of thickness, the arc length used for linking should be as short as possible.

### 2.3. Analysis of Fulfillment Problems

This section does not study the global winding method of tee pipes from the mathematical theory, but from the practical application studies how to use a CAD/CAM tool to fully cover the tee pipes. After the bandwidth display is realized in the CAD/CAM system, the fiber coverage area can be directly reflected. The designer can set up new design points to gradually cover the tee according to the coverage.

The method to fully cover the T-junction has some general rules. The most basic line type is shown in Figure 5a. For straight pipes, the function of the slip line coefficient is to change the winding angle of the boundary design point to 0° or 180° within the length of the straight pipe. The longer the straight pipe, the smaller the required slip line coefficient. If the maximum absolute value of the slip line coefficient is given, the extreme value of the winding angle of the boundary design point can be obtained, as seen in Equation (6). It can be seen that the extreme value of the winding angle of the boundary design point is only related to the extreme value of the aspect ratio l/R and the slip line coefficient λ. Therefore, when the length-to-diameter ratio of the main pipe and the branch pipe are different, the extreme value of the winding angle at the boundary design point is also different. If the winding angle of the boundary design point exceeds the limit of Equation (6), the trajectory cannot turn back within the length of the straight pipe. Therefore, when designing the winding track of the T-junction, it is necessary to consider the aspect ratio of each straight pipe to set the winding angle.
(6)$$\alpha_{0}=\left\{\begin{array}{l} \arccos\left(\frac{1}{(l/R)\cdot \lambda_{\max}+1}\right),\alpha_{0}<\frac{\pi }{2}\\ \arccos\left(\frac{1}{(l/R)\cdot \lambda_{\min}-1}\right),\alpha_{0}>\frac{\pi }{2} \end{array}\right.$$

In the equation, $\lambda_{\max}$, $\lambda_{\min}$—Maximum slippage factor and minimum slippage factor, $\lambda_{\max}=-\lambda_{\min}$; $\alpha_{0}$—Extreme value of winding angle of boundary design point (rad), $\alpha_{0}\in [$0, π/2) ∪ (π/2, π]; l—Length of straight pipe (mm);

A series of winding trajectories generated along L1 in Figure 5c have already covered most of the T-junction, leaving only the shoulders and parts of the frontal area not yet covered. After picking up new design points in the uncovered area using the CAD/CAM software to generate new winding trajectories and several iterations of the design, the T-junction is finally completed, as shown in Figure 5d.

The above new design points and winding direction generated by the trajectory on the T-junction on the boundary of the three straight pipes are used as input, and the algorithm in Section 2.1 is used to generate the trajectory on the straight pipe, as Figure 6a shows. In the picture, the uncovered area is scattered and symmetrically distributed on the straight pipes. If not relying on CAD/CAM graphic display, it is difficult to predict which areas are not covered in mathematical theory. This also proves the necessity of researching CAD/CAM technology for tee pipe winding.

In the CAD/CAM software developed in this paper, not only does it allow freely setting design points on T-junctions, but also on straight pipes. Figure 6a shows the process of inserting a new design point in the uncovered area on the straight pipe. The inserted trajectory generates a new design point on the boundary of the T-junction while a track occurs on the T-junction. Figure 6c shows the trajectory of the T-junction that is finally covered after the connection.

## 3. Post-Processing of Winding Trajectories

Generating winding trajectories is the first step in the winding process. This section aims to post-process the winding trajectories to generate the motion path in the tee pipe winding. The position of the yarn exit point is calculated by adding constraints, analyzing and discussing the interference situation in the winding motion, studying the reasonable obstacle avoidance algorithm and completing the calculation of the motion coordinates.

### 3.1. Method of Calculating the Yarn Exiting Points Position

The basic winding process is as shown in Figure 7. During the winding process, the doffing point keeps moving along the planned fiber trajectory and the connection line between the exit point and the doffing point coincides with the tangent of the fiber trajectory at the doffing point. As long as the length of the hanging yarn is determined, the position of the yarn exit point can be uniquely determined. In this paper, the yarn exit point is constrained on an envelope surface to determine the length of the suspended yarn.

The shape of the envelope is similar to that of the tee pipe, except that the main pipe and branch pipe are directly intersected instead of being connected by a toroidal transition, as shown in Figure 7b.

After determining the geometric size of the envelope surface, the position of the yarn exit point can be calculated. The position of the doffing point and the tangent vector have been obtained in the trajectory generation. Next, the intersection point position between the space ray and the envelope surface needs to be solved. The solution method is related to the form of the surface. Theoretically, after that the motion coordinates calculation is finished, but in actual winding process there are still interferences occurring.

The interferences in the winding process include the interference between the suspension yarn and the mold and the interference between the winding head and the mold. This section studies the interference caused by the position of the winding heads in the winding of tee pipes, analyzes the causes, and proposes their corresponding solutions.

### 3.2. Interference and Its Identification and Solution

#### 3.2.1. Interference between Molds and Winding Heads

In actual winding, the main pipe of tee pipes needs to be passed through a mandrel, and the chuck of the main shaft clamps the mandrel to drive the tee pipe mold to rotate. It is the existence of the mandrel that causes interferences. When the winding head moves to the two ends of the main pipe’s envelope surface, interferences occur when it is too close to the mandrel, as shown in Figure 7b.

In order to solve this kind of interference, this section sets a minimum circle C on the end surface. When the position of the yarn exit point is inside the minimum circle, the yarn exit point is translated along the radius of the minimum circle. The minimum circle radius depends on the size of the winding head and shaft and needs to ensure that no interference occurs. As shown in Figure 7d, *p*_1_ is the yarn exit point where interference occurs, and *p*_2_ is the yarn exit point after translation.

Except for the interference situation mentioned above, there is another type of interference caused by interpolation motion. After completing the calculation of the motion coordinates, the obtained yarn exit point is on the cylindrical surface, but the middle position between the yarn exit points is not on the cylindrical surface, which may cause interference between the winding equipment and the mold. As shown in Figure 7f, the positions of the two adjacent yarn exit points are *p*_1_ and *p*_2_ respectively. Therefore, this type of interference can be resolved by inserting intermediate points on the envelope surface in the post-processing. However, the premise of this method is that the distance between the doffing points cannot be too long, otherwise the newly inserted points may cause interference between the fiber and the mold.

#### 3.2.2. Interference between Molds and Fibers

During the winding process, there also are interferences between the fiber and the mold. This interference causes the winding track to deviate from the design track, resulting in slippage, overhead, and other phenomena, which seriously affect the winding quality, as shown in Figure 8. In this case, the reasonable method should be to go over the main pipe to ensure that the fiber can still be wound on the branch pipe, as shown in Figure 8. From the above analysis, it can be seen that for the interference between the fiber and the mold, spiral insertion can still be used, but the position of the doffing point must ensure that the yarn exit point after insertion cannot interfere with the straight tube. In fact, the post-processing method is required to select the spiral that does not cause interferences among the spirals in the two winding directions as the new path is inserted.

#### 3.2.3. Interference under Different Coordinate System

Four coordinate system

The spindle rotation coordinate of the four-coordinate winding rotation coordinate *θ*, the transverse coordinate x, the longitudinal coordinate z of the winding head, and the yarn exit roller rotation coordinate “β” are obtained by Equation (7).
(7)$$\left\{\begin{array}{l} \theta =\arctan\left(\frac{y_{0}}{z_{0}}\right)\\ x=x_{0}\\ z=\sqrt{y_{0}^{2}+z_{0}^{2}}\\ \beta =\arctan\left(-\frac{e_{y0}}{e_{y1}}\right) \end{array}\right.$$

The equations given in the Equation (7) do not take into account the influence of the diameter of the yarn exit roller, and directly equate the position of the yarn outlet point with the center position of the roller. When the mold is large and the yarn exit roller is far away from the mold, the error caused by this setting can be ignored. However, when the size of the mold is close to the size of the yarn exit roller, the interference shown in Figure 9a will occur. In order to avoid that, the posture of the yarn exit roller must be corrected from roller A to roller A’ in the post-processing, as shown in Figure 9c,d.

In summary, the revised four-axis winding coordinates are shown in Equation (8)
(8)$$\left\{\begin{array}{l} \theta =\arctan\left(\frac{y_{0}}{z_{0}}\right)+\arctan(\frac{d.y}{d.z})\\ x=x_{0}+d.x\\ z=\sqrt{{\left(y_{0}+d.y\right)}^{2}+{\left(z_{0}+d.z\right)}^{2}}\\ \beta =\arctan\left(\frac{e_{y0}}{e_{y2}\sin\Delta \theta -e_{y1}\cos\Delta \theta }\right) \end{array}\right.$$
where $d=\mathrm{normalize}(\mathrm{Mx}-\frac{\mathrm{Mx}\cdot \mathrm{Fy}}{{\left|\mathrm{Fy}\right|}^{2}}\mathrm{Fy})\cdot r$, is the offset vector from O_A_ to O_B_, in which normalize is the vector normalization function. and *d.y* and *d.z* are respectively the *y* component (mm) and *z* component (mm) of the offset vector and Δ*θ* is the correction amount of the rotating coordinate (rad).

2.Five coordinate system

Compared with the four-coordinate winding, the five-coordinate winding has a yaw coordinate of the winding head. When winding from the branch pipe to the T-junction, interferences inevitably occur, as shown in Figure 10a. The deflection amplitude of the winding head is too large, which leads to interferences. Therefore, interference can be avoided by limiting the range of yaw coordinates. This section discusses the five-coordinate solution under interference conditions. In the common five-axis winding equipment, the rotation coordinate of the yarn exit roller is attached to the yaw coordinate, as shown in Figure 10b.

Like the four-axis solution, the five-axis solution also takes into account the influence of the roller diameter. The Fz axis is tangent to the surface of the yarn exit roller. Spindle rotation causes the yarn exit roller to the XZ plane of the fixed coordinate O. The five coordinates are obtained by Equation (9).
(9)$$\left\{\begin{array}{l} \theta =\arctan\left(\frac{y_{0}+re_{z1}}{z_{0}+re_{z2}}\right)\\ x=x_{0}+re_{z0}\\ z=\sqrt{{\left(y_{0}+re_{z1}\right)}^{2}+{\left(z_{0}+re_{z2}\right)}^{2}}\\ \alpha =\arctan\left(-\frac{e_{x2}}{e_{x0}}\right)\\ \beta =\arctan\left(\frac{e_{x1}}{e_{x0}\cos\alpha -e_{x2}\sin\alpha }\right) \end{array}\right.$$

For the interference situation in Figure 9a, the interference can be prevented by setting the constraint yaw coordinates in [−α_abs_, α_abs_], as shown in Figure 9b. The five-coordinate solution with interference is similar to the four-coordinate solution. The rotation coordinate of the main shaft of the five-axis winding *θ*, the horizontal coordinate x, the longitudinal coordinate z, the yaw coordinate, and the yarn exit roller rotation coordinate β of the winding head in the case of interferences are obtained, as shown in Equation (10).
(10)$$\left\{\begin{array}{l} \theta =\arctan\left(\frac{y_{0}}{z_{0}}\right)+\arctan(\frac{d.y}{d.z})\\ x=x_{0}+d.x\\ z=\sqrt{{\left(y_{0}+d.y\right)}^{2}+{\left(z_{0}+d.z\right)}^{2}}\\ \alpha =\alpha_{\lim}\\ \beta =\arctan\left(\frac{\left(e_{y1}\sin\Delta \theta +e_{y2}\cos\Delta \theta \right)\sin\alpha -e_{y0}\cos\alpha }{e_{y1}\cos\Delta \theta -e_{y2}\sin\Delta \theta }\right) \end{array}\right.$$

It is worth mentioning that the interference solution proposed in this section modifies the trajectory of the yarn exit point so that the winding trajectory of the doffing point deviates from the design trajectory. In another word, for avoiding interferences, the offset of the yarn exit point causes the deviation of the doffing point at the beginning and ending of the trajectory for solving interferences. But after avoiding the interference, the yarn exit point goes back to the designed trajectory as well as the doffing point. Therefore, the trajectory still obeys the designed one after avoiding the interference. In the winding experiment, it is found that although the modified trajectory segment causes the deviation, the fiber first slip away from the designed position on the surface of the mold during the winding process, and then slide back to the designed trajectory. And as the winding progresses, the deviation of the trajectory becomes smaller and smaller, and the fibers can still be arranged closely together. Therefore, in actual engineering applications, the offset caused by the obstacle avoidance algorithm is acceptable.

## 4. Implementation Method

### 4.1. CAD/CAM Software Development

#### 4.1.1. Software Framework and User Interface

This section uses C++ language to develop a special CAD/CAM software for tee pipes ased on Qt framework, “FiberStudio”by Cheng Chang and Jihao Qin of Harbin Institute of Technology in Harbin city Heilongjiang province China. The structure of the software makes reference to the MVC pattern and separates the interface, data, and business logic. The software source code contains three major components: user interface, controller and data. The specific composition is shown in Figure 11a. The FiberStudio main user interface is shown in Figure 11b,c shows the simulation of the winding process.

#### 4.1.2. Fiber Display and Software Interaction Technique

The display of fiber bundles is quite special. There are two problems in the OpenGL display mode: first, the mold blocks a part of the fibers; second, the hierarchical relationship of the fiber bundles cannot be presented, as shown in Figure 12a. These two problems do not only affect the designer’s visual judgment, but also when the tracks gradually increase, it is difficult for the designer to judge whether the area is covered or not. The causes of these two problems are analyzed as well as the corresponding solutions are proposed.

The first question is the mold blocks a part of the fibers. The basic unit of graphic drawing in OpenGL is triangular facets, as shown in Figure 12b. The apex of the tee patch and the apex of the fiber patch are both located at the theoretical position of the tee pipe. But in OpenGL, the vertices of the patches are connected by planes, and intersections occur. From the observer’s perspective, the closer surface can be seen instead of the farther one. When the FiberStudio software builds the model, the tee pipe is assigned a smaller vertex sampling interval than the fiber bundle. Therefore, in Figure 12a, most of the fiber bundles are covered by the tee pipe instead of the tee pipe being covered by the fiber bundles. In order to solve this problem, the scaling factor of the transformation matrix from the model coordinate system to the world coordinate system is set to 0.99 in the vertex shader program of FiberStudio, that is, the size of the three-way patch is appropriately reduced. That avoids the cross between the tee pipe’s face sheet and the fiber face sheet, as shown in Figure 12b.

The second problem is caused by the mutual shielding of fiber bundles. The vertices of the different levels of the fiber patch are still at the theoretical position of the tee, and the left and right edges of the fiber bundle are also at the theoretical position of the tee. Therefore, the hierarchical relationship is not clear. In order to solve this problem, the drawing method of OpenGL is controlled in this article.

In the process of converting a three-dimensional data of the spatial vertex into a two-dimensional data of the image, OpenGL discards the occluded invisible surface, and only renders the output pixel value for the visible surface. Visible and invisible patches are judged by the depth value. The depth value in gl_Position is used to calculate the depth value of the fragment, and the depth value of the fragment is then used for the depth test. OpenGL maintains a depth buffer for each pixel in the image. In the depth test, if the depth value of a segment is less than the buffer value, the content of this pixel is replaced with the value of this segment, and the buffer value is updated.

In OpenGL, glDepthMask(GL_FALSE) and glDepethMask(GL_TRUE) control whether to update the depth buffer value. After reducing the size of the three-way patch, the drawn fiber bundle is shown in Figure 12c. It can be seen that the face sheet of the fiber bundle is not covered by the tee face sheet, which means that the depth value of the fiber bundle face sheet is less than the depth value of the corresponding tee face sheet, so this can always pass the depth test. However, different layers of fiber bundles still intersect, indicating that the top fiber bundles do not always pass the depth test. This is because the depth buffer value has been updated to the depth buffer value of the fiber bundle patch at this time. Therefore, FiberStudio calls glDepthMask (GL_FALSE) before starting to draw the fiber bundle to make the depth buffer read-only. This means that the drawing order of the fiber bundle determines the sequence of the fiber bundle, because the fiber bundle drawn later can always pass the depth test and cover the fiber bundle drawn first, as shown in Figure 12d.

The interaction technology is designed to help designers collaborate with software to quickly and accurately view, input, and output information. In FiberStudio, the main process is to interact with the mouse. The user can use the mouse to pan, rotate, and zoom the model, and pick up design points on the surface. In Section 2.3, the method of picking up new design points on the T junction and straight pipes with the mouse and adding tracks in the uncovered area is described. After entering the picking state, the click action of the mouse is processed in the GLWidget component of the software, and a new design point is created on the surface of the tee pipe.

The calculation process of generating 3D points according to the 2D screen coordinates when the mouse is clicked is related to the display principle of OpenGL. The projection method in FiberStudio is a perspective projection, as shown in Figure 4, Figure 5, Figure 6, Figure 7, Figure 8, Figure 9 and Figure 10. In this paper, the area restricted by the near plane, far plane and the camera’s angle of view is called the visible volume. OpenGL only renders the patches inside the visible volume, and renders the points of the rays starting from the center of the camera to the same screen coordinates. Taking Figure 12e as an example, P_1_, P_2_ and P_3_ are located on the same ray. After the vertex shader calculation, the x and y coordinates of gl_Position are the same while the z coordinate is different. The z coordinates of P_1_ and P_3_ are 0 and 1, respectively, and the z coordinates of P_2_ are between 0 and 1. The unProject function in the glm library can calculate the 3D point coordinates in the model coordinate system according to the screen coordinates, the depth value in the interval [0,1], the camera matrix, the model matrix, the perspective matrix and the viewport information. This article uses the unProject function to calculate the three-dimensional coordinates of P_1_ and P_3_ on the near and far planes respectively.

### 4.2. Winding Experiments

#### 4.2.1. Winding Experiments with Desktop Winding Machine

In this section, a winding experiment with a four-coordinate desktop winding machine is conducted to verify the correctness of the trajectory planning and post-processing. The experimental equipment is a small desktop winding machine, with spindle rotation, horizontal, vertical and yarn exit roller rotation coordinates. The three-way pipe used for winding is made of gypsum. The prepreg tape used for winding is an epoxy resin prepreg tape produced by Guangwei Composites Company. The mechanical properties of the prepreg tap are shown in Table 1.

When winding the branch pipe, the yarn exit roller continuously rotates at the same direction for multiple times. The traditional yarn feeding method causes the prepreg tape to be twisted and difficult to unfold. Therefore, the prepreg tape is wound on the material collection rod which is placed at the yarn outlet to rotate synchronously. The collection of prepreg liner paper is realized by the rotation of the yarn exit roller. The winding trajectory and process are shown in Figure 13a,c.

In order to improve the mechanical properties of the product, a vacuum bag curing process is adopted. After putting the vacuum bag into a hot air curing furnace, heating up to 120° with the furnace, holding it for 120 min, and then cooling it down to room temperature with the furnace, and the cured tee pipe and its microscope view of T-junction is shown in Figure 13b. The trajectory fully covered the tee pipe in the software as well as the cured tee pipe is covered.

#### 4.2.2. Winding Experiments with a Six-Axis Robot

Compared with machine tools, winding with robots has a higher degree of freedom and occupies smaller space. By replacing the end-effector, the robot can realize multiple processes of composite material forming. Therefore, the application of robots can significantly improve the automation level of composite manufacturing.

In order to avoid the twisting phenomenon of fiber when winding the branch pipe of tee pipes, a special winding head for robot winding tee pipes is designed in this paper, as shown in Figure 14a,c. There is a rotary motor attached to the winding head. The rotation of the motor is transmitted through the gear, shaft and synchronous belt, which drives the yarn reel and the guide to rotate synchronously. Thus the fiber twisting problem is solved. Figure 14b shows the robot winding system. The experiment is conducted using a 6 axies robot Comau NJ220-2.7, and Siemens 840D sl numerical control system as the control system. The winding head is installed at the end of the robot, and the tee pipe mold is clamped on the main shaft.

The tee pipe after winding is shown in Figure 14d. It is seen from the figure that the surface of the tee pipe has been fully covered with fibers, and the line is consistent with the planned trajectory. Except the end of straight pipes, the thickness of other parts is relatively even. The reason for that is most of the tracks retrace at the end of tee pipes, and the winding direction of the retrace point is perpendicular to the axis.

## 5. Conclusions

The paper presents a study of a general winding trajectory design and post-processing method for tee pipes. The works are as follows.

(1)A design method of “scheduling parts first and at last connecting all” is put forward to solve the coupling problem among trajectories. What’s more, the concept of design patches is suggested, which transforms the trajectory design process of the whole tee pipe to a trajectory design process on multiple design patches. The non-geodesic algorithms of three types of curved surfaces are discussed, in which a calculation test with the parameter of the torus model: *R* = 100, *r* = 20, u = π/4, V = π and *λ* = 0.1 is done, pointing out that the method using arc length parameter s as an independent variable is more stable than using *u* or *v* in non-geodesic differential equations. Based on the topological information of patches and boundaries, the non-geodesic trajectories on different patches are connected to realize the tee pipe winding trajectory design. It is also found due to the α is 90° at the turning points, the fiber accumulation is serious at the end of straight pipes.(2)In the post-processing of winding trajectories, the yarn exit point is constrained on the envelope surface to obtain the yarn exit point trajectory. The interferences under three different situations were discussed and analyzed and the corresponding solution methods are given.(3)In the aspect of the tee pipe winding CAD/CAM technology, a special CAD/CAM software for tee pipe winding was developed based on C++ and Qt frameworks. The problem of the hierarchical relationship of fibers is solved by controlling the OpenGL drawing process and the picking up design points process of interaction technique is given. Through those, the design method and visual system in software to achieve the fully coverage of tee pipes are realized. Meanwhile, a special robot winding head is designed to solve the twisting problem. Experiments with a desktop winding machine and a six-axis robot are carried out to verify the correctness of the trajectory design and post-processing methods.

In order to validate the fully coverage of tee pipes which is measured and ensured by FiberStudio and visual sight in this paper because of the limitation of devices, vision system tests and Nondestructive Testing (NDT) will be carried out in the future. With the movements of the robot to avoid interferences, optical measurement technology should be applied to determine the deviation of the actual position from the designed position. Moreover, the model of tee pipe fully coverage from an overall perspective shall be derived to improve the uniformity of the fiber thickness. The results of these researches should help to improve the winding trajectory design method and the software and hardware of winding machines.

## Figures and Tables

**Figure 1 materials-14-00847-f001:**
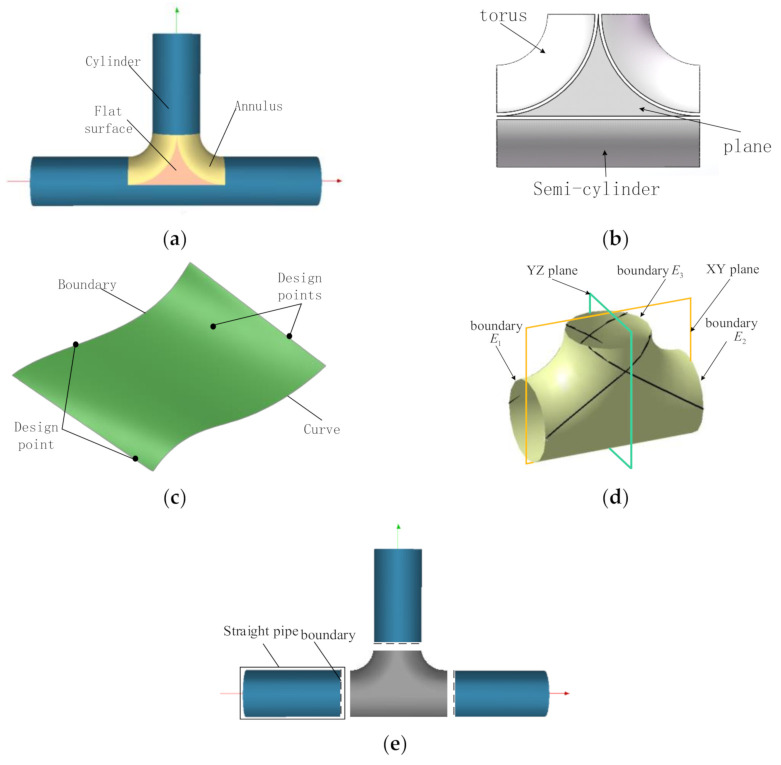
Schematics of the design method for illustration: (**a**) Tee pipes, (**b**) T-junction patch, (**c**) Design patches, (**d**) Symmetric trajectories, (**e**) Straight pipe boundary.

**Figure 2 materials-14-00847-f002:**
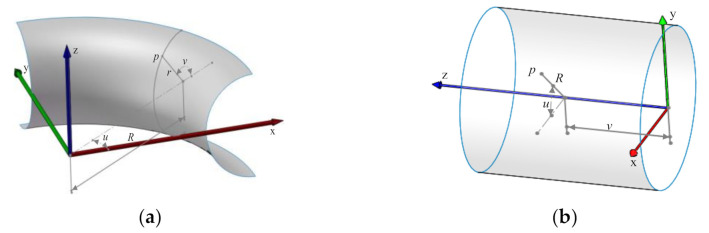
Coordinate system of different parts and compare between two trajectories with different variables. (**a**) Torus coordinate system; (**b**) Cylinder coordinate system; (**c**) Trajectory with u as an independent variable; (**d**) Trajectory with s as an independent variable.

**Figure 3 materials-14-00847-f003:**
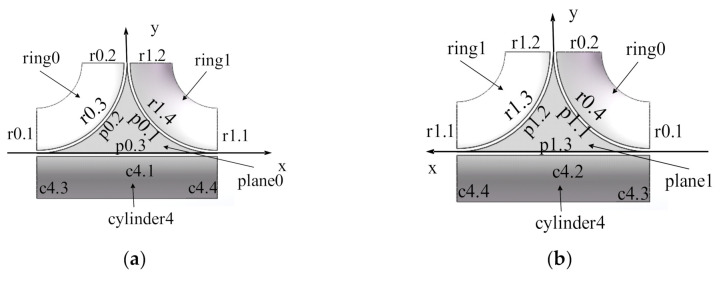
Schematics of T-junction patches: (**a**) Front view; (**b**) Postback view.

**Figure 4 materials-14-00847-f004:**
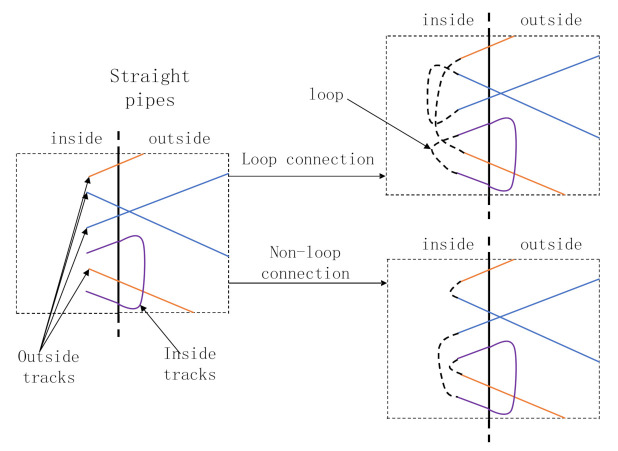
Schematic of loop connection.

**Figure 5 materials-14-00847-f005:**
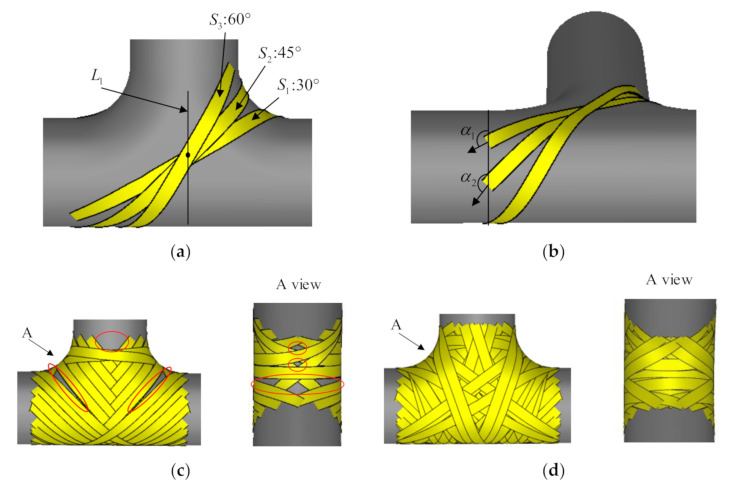
Full cover problems of T-junction: (**a**) Tracks corresponding to different winding angles; (**b**) Winding angle of boundary design point; (**c**) Uncovered area of T-junction; (**d**) Fully covered area of T-junction.

**Figure 6 materials-14-00847-f006:**
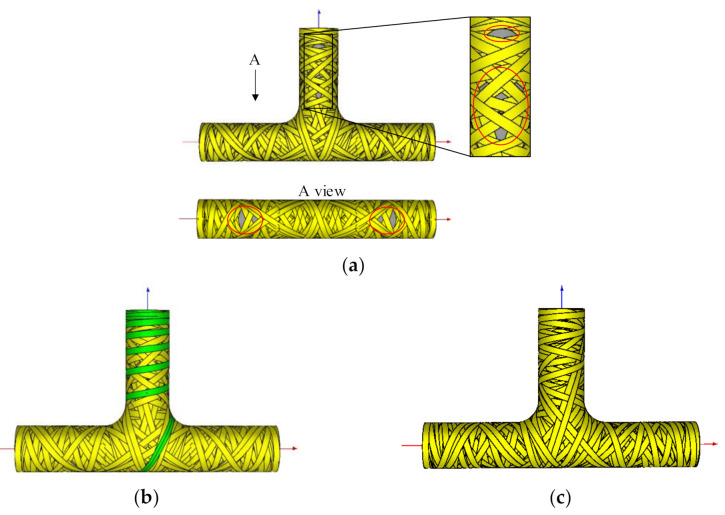
Schematics of fulfillment: (**a**) Trajectory on straight pipes derived from the trajectory on T-junction; (**b**) Inserted trajectory for straight pipes; (**c**) Trajectory after connection.

**Figure 7 materials-14-00847-f007:**
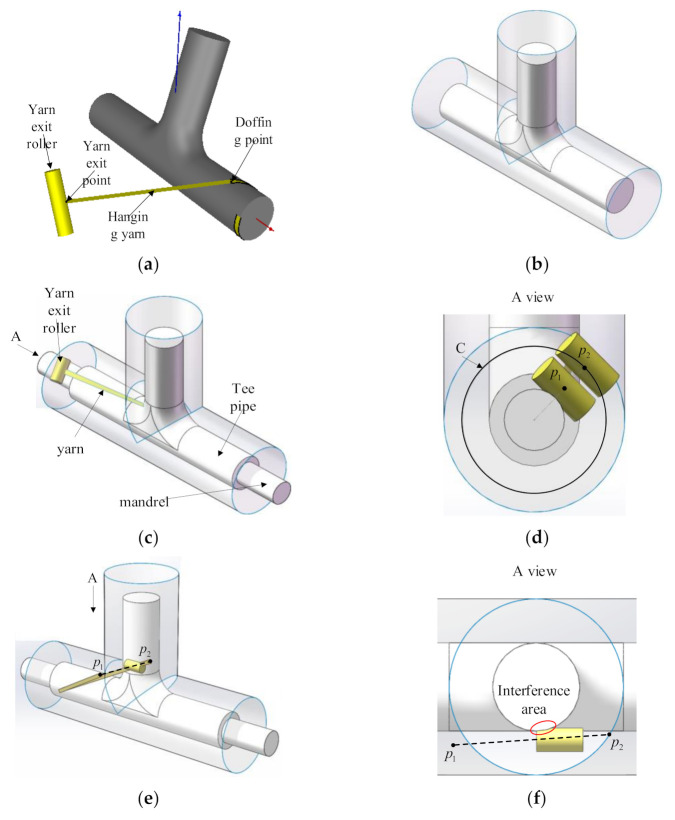
Schematics of interferences: (**a**) Schematic diagram of winding concept; (**b**) Envelop surface intersects directly; (**c**) Mandrel interferences; (**d**) Yarn exit offset; (**e**) Interference in the linear interpolation process; (**f**) Interference area.

**Figure 8 materials-14-00847-f008:**
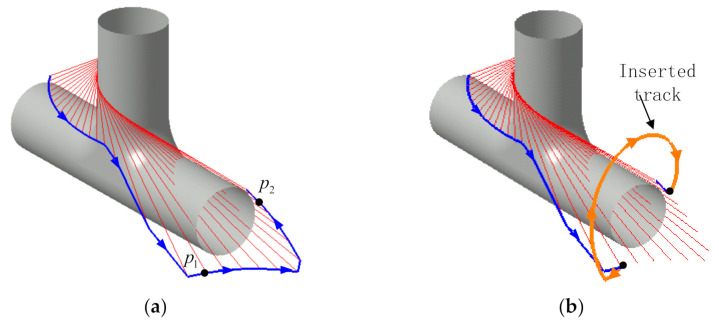
Schematic of the interference and solution: (**a**) Interference diagram; (**b**) Inserted track.

**Figure 9 materials-14-00847-f009:**
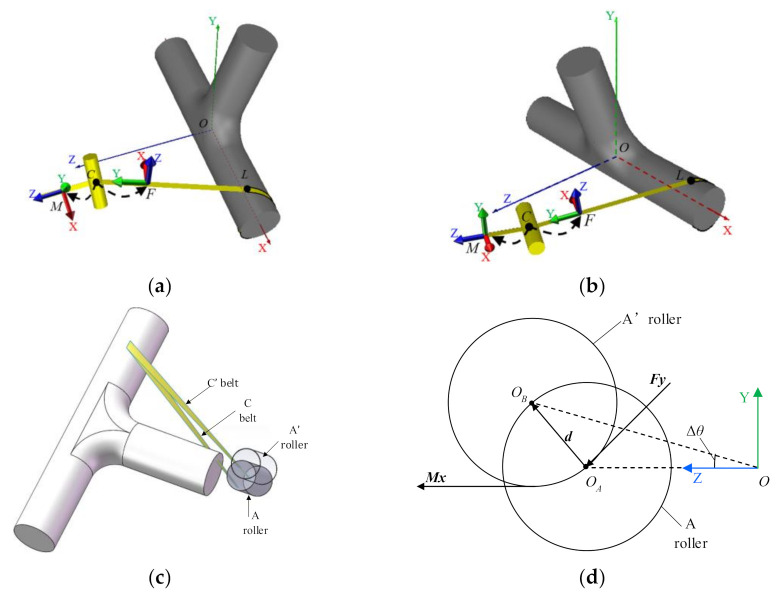
Interferences in four and five-coordinate system: (**a**) Four coordinate system; (**b**) Five coordinate system; (**c**) Hanging yarn corresponding to yarn exit positions; (**d**) Position correction.

**Figure 10 materials-14-00847-f010:**
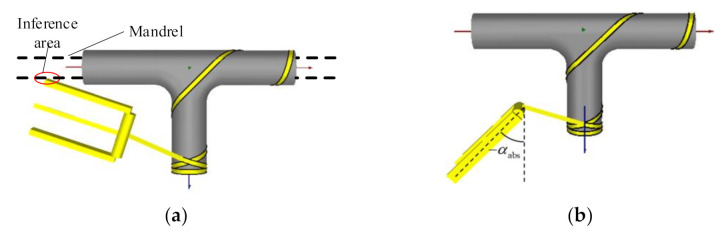
Interference in five coordinate system: (**a**) Interference caused by a large swing; (**b**) Restrained yaw coordinates.

**Figure 11 materials-14-00847-f011:**
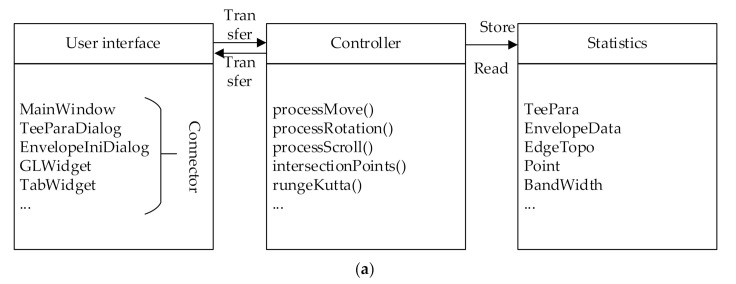
CAD/CAM software: (**a**) FiberStudio’s frame; (**b**) CAD/CAM user interface; (**c**) Simulation of the winding process.

**Figure 12 materials-14-00847-f012:**
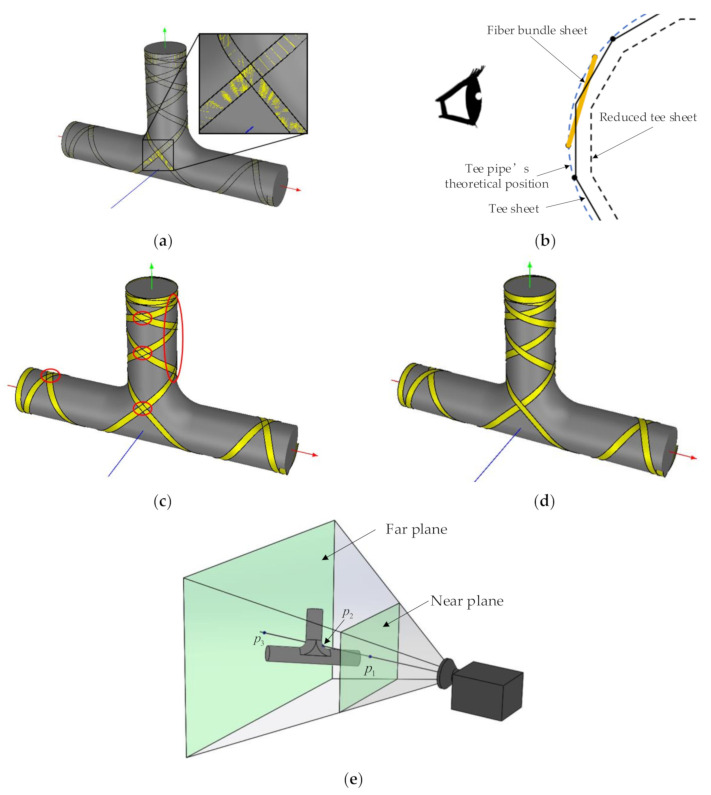
Fiber display and perspective model (**a**) Overlap among sheets; (**b**) Schematic of intersection; (**c**) Writable depth buffer; (**d**) Depth buffer read-only; (**e**) Perspective model.

**Figure 13 materials-14-00847-f013:**
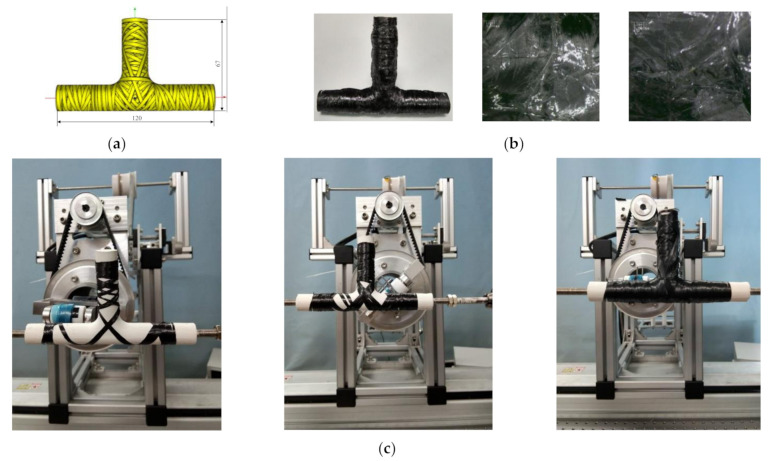
Winding experiment with a desktop winding machine (**a**) Designed trajectory by FiberStudio; (**b**) Cured tee pipe and its microscope view of T-junction; (**c**) Three different phases of the winding process.

**Figure 14 materials-14-00847-f014:**
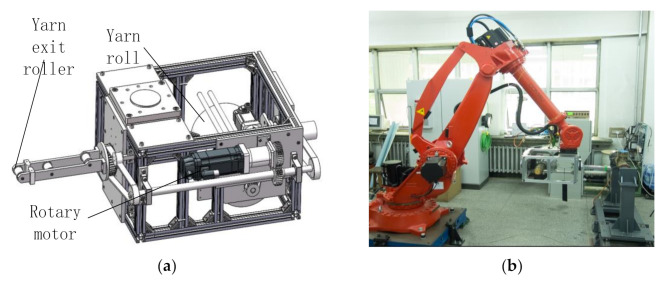
Six-axis robot winding system: (**a**) Winding head structure; (**b**) Six-axis robot; (**c**) Winding head; (**d**) Tee pipe after winding.

**Table 1 materials-14-00847-t001:** Mechanical properties of 6511 type carbon fiber/epoxy prepreg tape ^1^.

Mechanical Properties Parameter	Value	
Thermal Conductivity (J·s ^−1^·°C^−1^·mm^−1^)	2 × 10^−4^	
Density (kg·m^−3^)	1.49 × 10^3^	
Specific heat (J·kg^−1^·°C^−1^)	1.29 × 10^−9^	
Elastic Modulus (GPa)	E_1_ = 121	E_2_ = E_3_ = 4.7
Shear Modulus (GPa)	G_12_ = G_13_ = 4.7	G_23_ = 0.049
Poisson Ratio	ʋ_12_ = ʋ_13_ = 0.27	ʋ_23_ = 0.4

^1^ Part of mechanical properties are provided by Guangwei Composites Company.

## Data Availability

Data sharing not applicable.

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
