# Peer review of "A Non-Geodesic Trajectory Design Method and Its Post-Processing for Robotic Filament Winding of Composite Tee Pipes"

_materials, 2021, doi:10.3390/ma14040847_

Round 1

Reviewer 1 Report

This paper presents an approach to manufacturing tee pipes using the filament winding technique. A method for designing the winding pattern is developed by dividing the winding area into three patches and determining the trajectories for each patch before joining the patches to form a complete winding pattern. By selecting appropriate design points, complete coverage of the tee pipe is ensured. To avoid collisions and overlaps between winding paths, mold and winding head, the winding pattern is further optimized by inserting intermediate points and adjustment the position of the yarn exit roller. The software FiberStudio is used to calculate the movements of the six-axis robot and the winding head. To validate the winding pattern, a prototype is manufactured and visually evaluated.

A minor revision is required to publish the paper:

- Please consider the entire state of the art and discuss differences between your approach and the given ones:

Dackweiler, M., Mayer, T., Coutandin, S. et al. Modeling and optimization of winding paths to join lightweight profiles with continuous carbon fibers. Prod. Eng. Res. Devel. 13, 519-528 (2019). https://doi.org/10.1007/s11740-019-00914-2

- In Figure 1, you show schematically the design method. You might add the T junction patch with the labeled tori, the two planes and the semi-cylinder.

- To better understand equation 1, it might be helpful to highlight α in a figure

- In line 130 you give λ=0.1. Is this an estimate or is λ measured by experiment?

- In line 340 you mention a slippage of the fibers. Can you please describe this in more detail? What method/tool is used to measure the displacement due to slippage?

- Section 4.2 shows the results of the experiment. How is it determined that the line matches the planned trajectory? What measuring equipment is used?

Reviewer 2 Report

The present work deals with robotic filament winding technology property to produce complex shape parts. In particular, the authors study a novel non geodesic trajectory design method to get a continuous trajectory for tee pipe winding.

The topic of this work is relevant but innovation compared to other software and/or models in the literature is not evident; what is reported is only a case study! For example, what is the potential of the proposed method compared to the one on which the cadwind software was developed? (https://www.material.be/cadwind/intro/): In the introduction, specify better the INNOVATION OF WORK.

* Mechanical properties of composite materials (CFRP) are missing, please insert them in a table.

* In paragraph "Winding Experiments with Six-axis Robot” the autors write: It is seen from the figure that the surface of the tee pipe has been fully covered with fibers, and the line is consistent with the planned trajectory … “

For a correct experimental analysis, it is advisable to carry out an evaluation of product quality “layer by layer” by vision system and NDT (also after curing process)!  

* In paragraph "conclusion" insert quantitative data of obtained results

Round 2

Reviewer 2 Report

I am satisfied that the authors have adequately addressed the reviewer' comments. In my opinion the revised paper in its current form it is acceptable for publication in the Journal.